# Convolutional Gaussian Processes

**Mark van der Wilk**
Department of Engineering
University of Cambridge, UK
mv310@cam.ac.uk

**Carl Edward Rasmussen**
Department of Engineering
University of Cambridge, UK
cer54@cam.ac.uk

**James Hensman**
prowler.io
Cambridge, UK
james@prowler.io

## Abstract

We present a practical way of introducing convolutional structure into Gaussian processes, making them more suited to high-dimensional inputs like images. The main contribution of our work is the construction of an inter-domain inducing point approximation that is well-tailored to the convolutional kernel. This allows us to gain the generalisation benefit of a convolutional kernel, together with fast but accurate posterior inference. We investigate several variations of the convolutional kernel, and apply it to MNIST and CIFAR-10, where we obtain significant improvements over existing Gaussian process models. We also show how the marginal likelihood can be used to find an optimal weighting between convolutional and RBF kernels to further improve performance. This illustration of the usefulness of the marginal likelihood may help automate discovering architectures in larger models.

## 1   Introduction

Gaussian processes (GPs) [1] can be used as a flexible prior over functions, which makes them an elegant building block in Bayesian nonparametric models. In recent work, there has been much progress in addressing the computational issues preventing GPs from scaling to large problems [2, 3, 4, 5]. However, orthogonal to being able to algorithmically handle large quantities of data is the question of how to build GP models that generalise well. The properties of a GP prior, and hence its ability to generalise in a specific problem, are fully encoded by its covariance function (or kernel). Most common kernel functions rely on rather rudimentary and local metrics for generalisation, like the Euclidean distance. This has been widely criticised, notably by Bengio [6], who argued that deep architectures allow for more non-local generalisation. While deep architectures have seen enormous success in recent years, it is an interesting research question to investigate what kind of non-local generalisation structures *can* be encoded in shallow structures like kernels, while preserving the elegant properties of GPs.

Convolutional structures have non-local influence and have successfully been applied in neural networks to improve generalisation for image data [see e.g. 7, 8]. In this work, we investigate how Gaussian processes can be equipped with convolutional structures, together with accurate approximations that make them applicable in practice. A previous approach by Wilson et al. [9] transforms the inputs to a kernel using a convolutional neural network. This produces a valid kernel since applying a deterministic transformation to kernel inputs results in a valid kernel [see e.g. 1, 10], with the (many) parameters of the transformation becoming kernel hyperparameters. We stress that our approach is different in that the process itself is convolved, which does not require the introduction of additional parameters. Although our method does have inducing points that play a similar role to the filters in a convolutional neural network (convnet), these are variational parameters and are therefore more protected from over-fitting.

## 2  Background

Interest in Gaussian processes in the machine learning community started with the realisation that a shallow but infinitely wide neural network with Gaussian weights was a Gaussian process [11] – a nonparametric model with analytically tractable posteriors and marginal likelihoods. This gives two main desirable properties. Firstly, the posterior gives uncertainty estimates, which, combined with having an infinite number of basis functions, results in sensibly large uncertainties far from the data (see Quiñonero-Candela and Rasmussen [12, fig. 5] for a useful illustration). Secondly, the marginal likelihood can be used to select kernel hyperparameters. The main drawback is an $\mathcal{O}\left(N^3\right)$ computational cost for $N$ observations. Because of this, much attention over recent years has been devoted to scaling GP inference to large datasets through sparse approximations [2, 13, 14], minibatch-based optimisation [3], exploiting structure in the covariance matrix [e.g. 15] and Fourier methods [16, 17].

In this work, we adopt the variational framework for approximation in GP models, because it can simultaneously give a computational speed-up to $\mathcal{O}\left(NM^2\right)$ (with $M \ll N$) through sparse approximations [2] and approximate posteriors due to non-Gaussian likelihoods [18]. The variational choice is both elegant and practical: it can be shown that the variational objective minimises the KL divergence across the entire latent process [4, 19], which guarantees that the exact model will be approximated given enough resources. Other methods, such as EP/FITC [14, 20, 21, 22], can be seen as approximate models that do not share this property, leading to behaviour that would not be expected from the model that is to be approximated [23]. It is worth noting however, that our method for convolutional GPs is not specific to the variational framework, and can be used without modification with other objective functions, such as variations on EP.

### 2.1  Gaussian variational approximation

We adopt the popular choice of combining a sparse GP approximation with a Gaussian assumption, using a variational objective as introduced in [24]. We choose our model to be

$$f(\cdot)\,|\,\theta \sim \mathcal{GP}\left(0,\,k(\cdot,\cdot)\right), \tag{1}$$

$$y_i\,|\,f, \mathbf{x}_i \overset{iid}{\sim} p(y_i\,|\,f(\mathbf{x}_i)), \tag{2}$$

where $p(y_i\,|\,f(\mathbf{x}_i))$ is some non-Gaussian likelihood, for example a Bernoulli distribution through a probit link function for classification. The kernel parameters $\theta$ are to be estimated by approximate maximum likelihood, and we drop them from the notation hereon. Following Titsias [2], we choose the approximate posterior to be a GP with its marginal distribution specified at $M$ "inducing inputs" $\mathbf{Z} = \{\mathbf{z}_m\}_{m=1}^M$. Denoting the value of the GP at those points as $\mathbf{u} = \{f(\mathbf{z}_m)\}_{m=1}^M$, the approximate posterior process is constructed from the specified marginal and the prior conditional[1]:

$$\mathbf{u} \sim \mathcal{N}\left(\mathbf{m},\,\mathbf{S}\right), \tag{3}$$

$$f(\cdot)\,|\,\mathbf{u} \sim \mathcal{GP}\left(\mathbf{k_u}(\cdot)^\top \mathbf{K_{uu}^{-1}}\mathbf{u},\, k(\cdot,\cdot) - \mathbf{k_u}(\cdot)^\top \mathbf{K_{uu}^{-1}}\mathbf{k_u}(\cdot)\right). \tag{4}$$

The vector-valued function $\mathbf{k_u}(\cdot)$ gives the covariance between $\mathbf{u}$ and the remainder of $f$, and is constructed from the kernel: $\mathbf{k_u}(\cdot) = [k(\mathbf{z}_m, \cdot)]_{m=1}^M$. The matrix $\mathbf{K_{uu}}$ is the prior covariance of $\mathbf{u}$. The variational parameters $\mathbf{m}$, $\mathbf{S}$ and $\mathbf{Z}$ are then optimised with respect to the evidence lower bound (ELBO):

$$\mathrm{ELBO} = \sum_i \mathbb{E}_{q(f(\mathbf{x}_i))}\left[\log p(y_i\,|\,f(\mathbf{x}_i))\right] - \mathrm{KL}[q(\mathbf{u})||p(\mathbf{u})]. \tag{5}$$

Here, $q(\mathbf{u})$ is the density of $\mathbf{u}$ associated with equation (3), and $p(\mathbf{u})$ is the prior density from (1). Expectations are taken with respect to the marginals of the posterior approximation, given by

$$q(f(\mathbf{x}_i)) = \mathcal{N}\left(\mu_i, \sigma_i^2\right), \tag{6}$$

$$\mu_i = \mathbf{k_u}(\mathbf{x}_i)^\top \mathbf{K_{uu}^{-1}}\mathbf{m}, \tag{7}$$

$$\sigma_i^2 = k(\mathbf{x}_i, \mathbf{x}_i) + \mathbf{K_{fu}}\mathbf{K_{uu}^{-1}}(\mathbf{S} - \mathbf{K_{uu}})\mathbf{K_{uu}^{-1}}\mathbf{K_{uf}}. \tag{8}$$

The matrices $\mathbf{K_{uu}}$ and $\mathbf{K_{fu}}$ are obtained by evaluating the kernel as $k(\mathbf{z}_m, \mathbf{z}_{m'})$ and $k(\mathbf{x}_n, \mathbf{z}_m)$ respectively. The KL divergence term of the ELBO is analytically tractable, whilst the expectation term can be computed using one-dimensional quadrature. The form of the ELBO means that stochastic optimisation using minibatches is applicable. A full discussion of the methodology is given by Matthews [19]. We optimise the ELBO instead of the marginal likelihood to find the hyperparameters.

## 2.2 Inter-domain variational GPs

Inter-domain Gaussian processes [25] work by replacing the variables $\mathbf{u}$, which we have above assumed to be observations of the function at the inducing inputs $\mathbf{Z}$, with more complicated variables made by some linear operator on the function. Using linear operators ensures that the inducing variables $\mathbf{u}$ are still jointly Gaussian with the other points on the GP. Implementing inter-domain inducing variables can therefore be a drop-in replacement to inducing points, requiring only that the appropriate (cross-)covariances $\mathbf{K_{fu}}$ and $\mathbf{K_{uu}}$ are used.

The key advantage of the inter-domain approach is that the approximate posterior mean's (7) effective basis functions $\mathbf{k_u}(\cdot)$ can be manipulated by the linear operator which constructs $\mathbf{u}$. This can make the approximation more flexible, or give other computational benefits. For example, Hensman et al. [17] used the Fourier transform to construct $\mathbf{u}$ such that the $\mathbf{K_{uu}}$ matrix becomes easier to invert.

Inter-domain inducing variables are usually constructed using a weighted integral of the GP:

$$u_m = \int \phi(\mathbf{x}; \mathbf{z}_m) f(\mathbf{x}) \, d\mathbf{x}, \tag{9}$$

where the weighting function $\phi$ depends on some parameters $\mathbf{z}_m$. The covariance between the inducing variable $u_m$ and a point on the function is then

$$\mathrm{cov}(u_m, f(\mathbf{x}_n)) = k(\mathbf{z}_m, \mathbf{x}_n) = \int \phi(\mathbf{x}; \mathbf{z}_m) k(\mathbf{x}, \mathbf{x}_n) \, d\mathbf{x}, \tag{10}$$

and the covariance between two inducing variables is

$$\mathrm{cov}(u_m, u_{m'}) = k(\mathbf{z}_m, \mathbf{z}_{m'}) = \iint \phi(\mathbf{x}; \mathbf{z}_m) \phi(\mathbf{x}'; \mathbf{z}_{m'}) k(\mathbf{x}, \mathbf{x}') \, d\mathbf{x} \, d\mathbf{x}'. \tag{11}$$

Using inter-domain inducing variables in the variational framework is straightforward if the above integrals are tractable. The results are substituted for the kernel evaluations in equations (7) and (8).

Our proposed method will be an inter-domain approximation in the sense that the inducing input space is different from the input space of the kernel. However, instead of relying on an integral transformation of the GP, we construct the inducing variables $\mathbf{u}$ alongside the new kernel such that the effective basis functions contain a convolution operation.

## 2.3 Additive GPs

We would like to draw attention to previously studied additive models [26, 27], in order to highlight the similarity with the convolutional kernels we will introduce later. Additive models construct a prior GP as a sum of functions over subsets of the input dimensions, resulting in a kernel with the same additive structure. For example, summing over each input dimension $i$, we get

$$f(\mathbf{x}) = \sum_i f_i(\mathbf{x}[i]) \implies k(\mathbf{x}, \mathbf{x}') = \sum_i k_i(\mathbf{x}[i], \mathbf{x}'[i]). \tag{12}$$

This kernel exhibits some non-local generalisation, as the relative function values along one dimension will be the same regardless of the input along other dimensions. In practice, this specific additive model is rather too restrictive to fit data well, since it assumes that all variables affect the response $\mathbf{y}$ independently. At the other extreme, the popular squared exponential kernel allows interactions between all dimensions, but this turns out to be not restrictive enough: for high-dimensional problems we need to impose *some* restriction on the form of the function.

In this work, we build an additive kernel inspired by the convolution operator found in convnets. The same function is applied to patches from the input, which allows adjacent pixels to interact, but imposes an additive structure otherwise.

# 3 Convolutional Gaussian Processes

We begin by constructing the exact convolutional Gaussian process model, highlighting its connections to existing neural network models, and challenges in performing inference.

**Convolutional kernel construction** Our aim is to construct a GP prior on functions on images of size $D = W \times H$ to real valued responses: $f : \mathbb{R}^D \to \mathbb{R}$. We start with a *patch-response function*, $g : \mathbb{R}^E \to \mathbb{R}$, mapping from patches of size $E$. We use a stride of 1 to extract all patches, so for patches of size $E = w \times h$, we get a total of $P = (W - w + 1) \times (H - h + 1)$ patches. We can start by simply making the overall function $f$ the sum of all patch responses. If $g(\cdot)$ is given a GP prior, a GP prior will also be induced on $f(\cdot)$:

$$g \sim \mathcal{GP}\left(0, k_g(\mathbf{z}, \mathbf{z}')\right) , \quad f(\mathbf{x}) = \sum_p g\left(\mathbf{x}^{[p]}\right) , \tag{13}$$

$$\implies f \sim \mathcal{GP}\left(0, \sum_{p=1}^{P} \sum_{p'=1}^{P} k_g\left(\mathbf{x}^{[p]}, \mathbf{x}'^{[p']}\right)\right) , \tag{14}$$

where $\mathbf{x}^{[p]}$ indicates the $p^{\text{th}}$ patch of the image $\mathbf{x}$. This construction is reminiscent of the additive models discussed earlier, since a function is applied to subsets of the input. However, in this case, the *same* function $g(\cdot)$ is applied to all input subsets. This allows all patches in the image to inform the value of the patch-response function, regardless of their location.

**Comparison to convnets** This approach is similar in spirit to convnets. Both methods start with a function that is applied to each patch. In the construction above, we introduce a single patch-response function $g(\cdot)$ that is non-linear and nonparametric. Convnets, on the other hand, rely on many linear filters, followed by a non-linearity. The flexibility of a single convolutional layer is controlled by the number of filters, while depth is important in order to allow for enough non-linearity. In our case, adding more non-linear filters to the construction of $f(\cdot)$ does not increase the capacity to learn. The patch responses of the multiple filters would be summed, resulting in simply a summed kernel for the prior over $g$.

**Computational issues** Similar kernels have been proposed in various forms [28, 29], but have never been applied directly in GPs, probably due to the prohibitive costs. Direct implementation of a GP using $k_f$ would be infeasible not only due to the usual cubic cost w.r.t. the number of data points, but also due to it requiring $P^2$ evaluations of $k_g$ *per element of* $\mathbf{K_{ff}}$. For MNIST with patches of size 5, $P^2 \approx 3.3 \cdot 10^5$, resulting in the kernel evaluations becoming a significant bottleneck. Sparse inducing point methods require $M^2 + NM$ kernel evaluations of $k_f$. As an illustration, the $\mathbf{K_{uu}}$ matrix for 750 inducing points (which we use in our experiments) would require $\sim 700$ GB of memory for backpropagation. Luckily, this can largely be avoided.

# 4 Inducing patch approximations

In the next few sections, we will introduce several variants of the convolutional Gaussian process, and illustrate their properties using toy and real datasets. Our main contribution is showing that convolutional structure can be embedded in kernels, and that they can be used within the framework of nonparametric Gaussian process approximations. We do so by constructing the kernel in tandem with a suitable domain in which to place the inducing variables. Implementation[2] requires minimal changes to existing implementations of sparse variational GP inference, and can leverage GPU implementations of convolution operations (see appendix). In the appendix we also describe how the same inference method can be applied to kernels with general invariances.

## 4.1 Translation invariant convolutional GP

Here we introduce the simplest version of our method. We start with the construction from section 3, with an RBF kernel for $k_g$. In order to obtain a tractable method, we want to approximate the

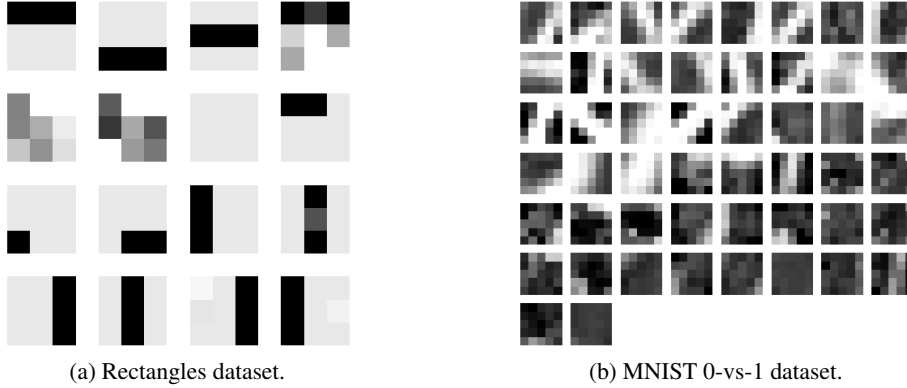

| (a) Rectangles dataset. | (b) MNIST 0-vs-1 dataset. |

Figure 1: The optimised inducing patches for the translation invariant kernel. The inducing patches are sorted by the value of their corresponding inducing output, illustrating the evidence each patch has in favour of a class.

true posterior using a small set of inducing points. The main idea is to place these inducing points in the input space of *patches*, rather than images. This corresponds to using inter-domain inducing points. In order to use this approximation we simply need to find the appropriate inter-domain (cross-) covariances $\mathbf{K_{uu}}$ and $\mathbf{K_{fu}}$, which are easily found from the construction of the convolutional kernel in equation 14:

$$k_{fu}(\mathbf{x}, \mathbf{z}) = \mathbb{E}_g \left[ f(\mathbf{x})g(\mathbf{z}) \right] = \mathbb{E}_g \left[ \sum_p g(\mathbf{x}^{[p]})g(\mathbf{z}) \right] = \sum_p k_g \left( \mathbf{x}^{[p]}, \mathbf{z} \right), \tag{15}$$

$$k_{uu}(\mathbf{z}, \mathbf{z}') = \mathbb{E}_g \left[ g(\mathbf{z})g(\mathbf{z}') \right] = k_g(\mathbf{z}, \mathbf{z}'). \tag{16}$$

This improves on the computation from the standard inducing point method, since only covariances between the image patches and inducing patches are needed, allowing $\mathbf{K_{fu}}$ to be calculated with $NMP$ instead of $NMP^2$ kernel evaluations. Since $\mathbf{K_{uu}}$ now only requires the covariances between *inducing patches*, its cost is $M^2$ instead of $M^2P^2$ evaluations. However, evaluating $\text{diag}\left[\mathbf{K_{ff}}\right]$ does still require $NP^2$ evaluations, although $N$ can be small when using minibatch optimisation. This brings the cost of computing the kernel matrices down significantly compared to the $\mathcal{O}\left(NM^2\right)$ cost of the calculation of the ELBO.

In order to highlight the capabilities of the new kernel, we now consider two toy tasks: classifying rectangles and distinguishing zeros from ones in MNIST.

**Toy demo: rectangles** The rectangles dataset is an artificial dataset containing 1200 images of size $28 \times 28$. Each image contains the outline of a randomly generated rectangle, and is labelled according to whether the rectangle has larger width or length. Despite its simplicity, the dataset is tricky for standard kernel-based methods, including Gaussian processes, because of the high dimensionality of the input, and the strong dependence of the label on multiple pixel locations.

To tackle the rectangles dataset with the convolutional GP, we used a patch size of $3 \times 3$ and 16 inducing points initialised with uniform random noise. We optimised using Adam [31] (0.01 learning rate & 100 data points per minibatch) and obtained $1.4\%$ error and a negative log predictive probability (nlpp) of $0.055$ on the test set. For comparison, an RBF kernel with 1200 optimally placed inducing points, optimised with BFGS, gave $5.0\%$ error and an nlpp of $0.258$. Our model is both better in terms of performance, and uses fewer inducing points. The model works because it is able to recognise and count vertical and horizontal bars in the patches. The locations of the inducing points quickly recognise the horizontal and vertical lines in the images – see Figure 1a.

**Illustration: Zeros vs ones MNIST** We perform a similar experiment for classifying MNIST 0 and 1 digits. This time, we initialise using patches from the training data and use 50 inducing features, shown in figure 1b. Features in the top left are in favour of classifying a zero, and tend to be diagonal or bent lines, while features for ones tend to be blank space or vertical lines. We get $0.3\%$ error.

**Full MNIST** Next, we turn to the full multi-class MNIST dataset. Our setup follows Hensman et al. [5], with 10 independent latent GPs using the same convolutional kernel, and constraining $q(\mathbf{u})$ to a Gaussian (see section 2). It seems that this translation invariant kernel is too restrictive for this task, since the error rate converges at around $2.1\%$, compared to $1.9\%$ for the RBF kernel.

## 4.2 Weighted convolutional kernels

We saw in the previous section that although the translation invariant kernel excelled at the rectangles task, it under-performed compared to the RBF on MNIST. Full translation invariance is too strong a constraint, which makes intuitive sense for image classification, as the same feature in different locations of the image can imply different classes. This can be remedied without leaving the family of Gaussian processes by relaxing the constraint of requiring each patch to give the same contribution, regardless of its position in the image. We do so by introducing a weight for each patch. Denoting again the underlying patch-based GP as $g$, the image-based GP $f$ is given by

$$f(\mathbf{x}) = \sum_p w_p g(\mathbf{x}^{[p]}) . \tag{17}$$

The weights $\{w_p\}_{p=1}^P$ adjust the relative importance of the response for each location in the image. Only $k_f$ and $k_{fu}$ differ from the invariant case, and can be found to be:

$$k_f(\mathbf{x}, \mathbf{x}) = \sum_{pq} w_p w_q k_g(\mathbf{x}^{[p]}, \mathbf{x}_q) , \tag{18}$$

$$k_{fu}(\mathbf{x}, \mathbf{z}) = \sum_p w_p k_g(\mathbf{x}^{[p]}, \mathbf{z}) . \tag{19}$$

The patch weights $\mathbf{w} \in \mathbb{R}^P$ are now kernel hyperparameters, and we optimise them with respect the the ELBO in the same fashion as the underlying parameters of the kernel $k_g$. This introduces $P$ hyperparameters into the kernel – slightly less than the number of input pixels, which is how many hyperparameters an automatic relevance determination kernel would have.

**Toy demo: rectangles** The errors in the previous section were caused by rectangles along the edge of the image, which contained bars which only contribute once to the classification score. Bars in the centre contribute to multiple patches. The weighting allows some up-weighting of patches along the edge. This results in near-perfect classification, with no classification errors and an nlpp of $0.005$.

**Full MNIST** The weighting causes a significant reduction in error over the translation invariant and RBF kernels (table 1 & figure 2). The weighted convolutional kernel obtains $1.22\%$ error – a significant improvement over $1.9\%$ for the RBF kernel [5]. Krauth et al. [32] report $1.55\%$ error using an RBF kernel, but using a leave-one-out objective for finding the hyperparameters.

## 4.3 Does convolution capture everything?

As discussed earlier, the additive nature of the convolutional kernel places constraints on the possible functions in the prior. While these constraints have been shown to be useful for classifying MNIST, we lose the guarantee (that e.g. the RBF provides) of being able to model any continuous function arbitrarily well in the large-data limit. This is because convolutional kernels are not universal [33, 34] in the image input space, despite being nonparametric. This places convolutional kernels in a middle ground between parametric and universal kernels (see the appendix for a discussion). A kernel that *is* universal *and* has some amount of convolutional structure can be obtained by summing an RBF component: $k(\mathbf{x}, \mathbf{x}') = k_{\text{rbf}}(\mathbf{x}, \mathbf{x}') + k_{\text{conv}}(\mathbf{x}, \mathbf{x}')$. Equivalently, the GP is constructed by the sum $f(\mathbf{x}) = f_{\text{conv}}(\mathbf{x}) + f_{\text{rbf}}(\mathbf{x})$. This allows the universal RBF to model any residuals that the convolutional structure cannot explain. We use the marginal likelihood estimate to automatically weigh how much of the process should be explained by each of the components, in the same way as is done in other additive models [27, 35].

Inference in such a model is straightforward under the usual inducing point framework – it only requires evaluating the sum of kernels. The case considered here is more complicated since we want the inducing inputs for the RBF to lie in the space of images, while we want to use inducing patches

for the convolutional kernel. This forces us to use a slightly different form for the approximating GP, representing the inducing inputs and outputs separately, as

$$\begin{bmatrix} \mathbf{u}_{\text{conv}} \\ \mathbf{u}_{\text{rbf}} \end{bmatrix} \sim \mathcal{N}\left( \begin{bmatrix} \boldsymbol{\mu}_{\text{conv}} \\ \boldsymbol{\mu}_{\text{rbf}} \end{bmatrix}, \mathbf{S} \right), \tag{20}$$

$$f(\cdot) \mid \mathbf{u} = f_{\text{conv}}(\cdot) \mid \mathbf{u}_{\text{conv}} + f_{\text{rbf}}(\cdot) \mid \mathbf{u}_{\text{rbf}}. \tag{21}$$

The variational lower bound changes only through the equations (7) and (8), which must now contain contributions of the two component Gaussian processes. If covariances in the posterior between $f_{\text{conv}}$ and $f_{\text{rbf}}$ are to be allowed, $\mathbf{S}$ must be a full-rank $2M \times 2M$ matrix. A mean-field approximation can be chosen as well, in which case $\mathbf{S}$ can be $M \times M$ block-diagonal, saving some parameters. Note that regardless of which approach is chosen, the largest matrix to be inverted is still $M \times M$, as $\mathbf{u}_{\text{conv}}$ and $\mathbf{u}_{\text{rbf}}$ are independent in the prior (see the appendix for more details).

**Full MNIST** By adding an RBF component, we indeed get an extra reduction in error and nlpp from $1.22\%$ to $1.17\%$ and $0.048$ to $0.039$ respectively (table 1 & figure 2). The variances for the convolutional and RBF kernels are $14.3$ and $0.011$ respectively, showing that the convolutional kernel explains most of the variance in the data.

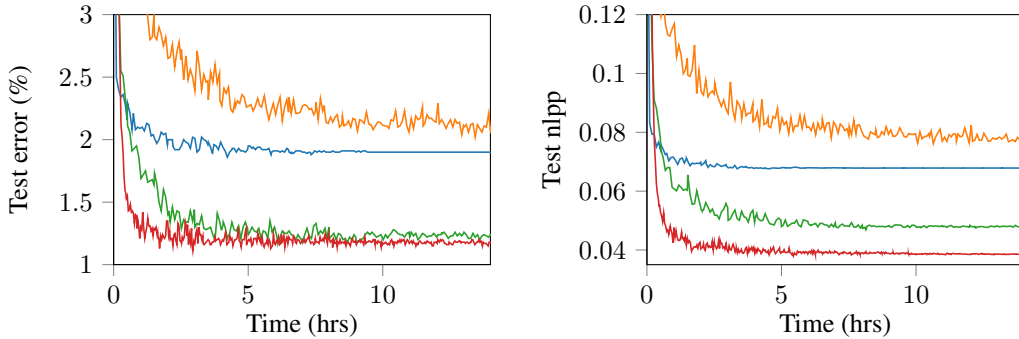

Figure 2: Test error (left) and negative log predictive probability (nlpp, right) for MNIST, using RBF (blue), translation invariant convolutional (orange), weighted convolutional (green) and weighted convolutional + RBF (red) kernels.

| Kernel | M | Error (%) | NLPP |
|---|---|---|---|
| Invariant | 750 | 2.08% | 0.077 |
| RBF | 750 | 1.90% | 0.068 |
| Weighted | 750 | 1.22% | 0.048 |
| Weighted + RBF | 750 | 1.17% | 0.039 |

Table 1: Final results for MNIST.

## 4.4 Convolutional kernels for colour images

Our final variants of the convolutional kernel handle images with multiple colour channels. The addition of colour presents an interesting modelling challenge, as the input dimensionality increases significantly, with a large amount of redundant information. As a baseline, the weighted convolutional kernel from section 4.2 can be used by taking all patches from each colour channel together, resulting in $C$ times more patches, where $C$ is the number of colour channels. This kernel can only account for linear interactions between colour channels through the weights, and is also constrained to give the same patch response regardless of the colour channel. A step up in flexibility would be to define $g(\cdot)$ to take a $w \times h \times C$ patch with all $C$ colour channels. This trades off increasing the dimensionality of the patch-response function input with allowing it to learn non-linear interactions between the colour channels. We call this the *colour-patch variant*. A middle ground that does not increase the dimensionality as much, is to use a different patch-response function $g_c(\cdot)$ for each colour channel.

We will refer to this as the *multi-channel* convolutional kernel. We construct the overall function $f$ as

$$f(\mathbf{x}) = \sum_{p=1}^{P} \sum_{c=1}^{C} w_{pc} g_c \left( \mathbf{x}^{[pc]} \right) . \tag{22}$$

For this variant, inference becomes similar to section 4.3, although for a different reason. While all $g_c(\cdot)$s can use the same inducing patch inputs, we need access to each $g_c(\mathbf{x}^{[pc]})$ separately in order to fully specify $f(\mathbf{x})$. This causes us to require separate inducing outputs for each $g_c$. In our approximation, we share the inducing inputs, while, as was done in section 4.3, representing the inducing outputs separately. The equations for $f(\cdot)|\mathbf{u}$ are changed only through the matrices $\mathbf{K_{fu}}$ and $\mathbf{K_{uu}}$ being $N \times MC$ and $MC \times MC$ respectively. Given that the $g_c(\cdot)$ are independent in the prior, and the inducing inputs are constrained to be the same, $\mathbf{K_{uu}}$ is a block-diagonal repetition of $k_g(\mathbf{z}_m, \mathbf{z}_{m'})$. All the elements of $\mathbf{K_{fu}}$ are given by

$$k_{fg_c}(\mathbf{x}, \mathbf{z}) = \mathbb{E}_{\{g_c\}_{c=1}^{C}} \left[ \sum_p w_{pc} g_c \left( \mathbf{x}^{[pc]} \right) g_c(\mathbf{z}) \right] = \sum_p w_{pc} k_g(\mathbf{x}^{[pc]}, \mathbf{z}) . \tag{23}$$

As in section 4.3, we have the choice to represent a full $CM \times CM$ covariance matrix for all inducing variables $\mathbf{u}$, or go for a mean-field approximation requiring only $C$ $M \times M$ matrices. Again, both versions require no expensive matrix operations larger than $M \times M$ (see appendix).

Finally, a simplification can be made in order to avoid representing $C$ patch-response functions. If the weighting of each of the colour channels is constant w.r.t. the patch location (i.e. $w_{pc} = w_p w_c$), the model is equivalent to using a patch-response function with an additive kernel:

$$f(\mathbf{x}) = \sum_p w_p \sum_c w_c g_c(\mathbf{x}^{[pc]}) = \sum_p w_p \tilde{g}(\mathbf{x}^{[pc]}) , \tag{24}$$

$$\tilde{g}(\cdot) \sim \mathcal{GP} \left( 0, \sum_c w_c k_c(\cdot, \cdot) \right) . \tag{25}$$

**CIFAR-10**   We conclude the experiments by an investigation of CIFAR-10 [36], where $32 \times 32$ sized RGB images are to be classified. We use a similar setup to the previous MNIST experiments, by using $5 \times 5$ patches. Again, all latent functions share the same kernel for the prior, including the patch weights. We compare an RBF kernel to 4 variants of the convolutional kernel: the baseline "weighted", the colour-patch, the colour-patch variant with additive structure (equation 24), and the multi-channel with mean-field inference. All models use 1000 inducing inputs and are trained using Adam. Due to memory constraints on the GPU, a minibatch size of 40 had to be used for the weighted, additive and multi-channel models.

Test errors and nlpps during training are shown in figure 3. Any convolutional structure significantly improves classification performance, with colour interactions seeming particularly important, as the best performing model is the multi-channel GP. The final error rate of the multi-channel kernel was $35.4\%$, compared to $48.6\%$ for the RBF kernel. While we acknowledge that this is far from state of the art using deep nets, it is a significant improvement over existing Gaussian process models, including the $44.95\%$ error reported by Krauth et al. [32], where an RBF kernel was used together with their leave-one-out objective for the hyperparameters. This improvement is orthogonal to the use of a new kernel.

## 5   Conclusion

We introduced a method for efficiently using convolutional structure in Gaussian processes, akin to how it has been used in neural nets. Our main contribution is showing how placing the inducing inputs in the space of patches gives rise to a natural *inter-domain* approximation that fits in sparse GP approximation frameworks. We discuss several variations of convolutional kernels and show how they can be used to push the performance of Gaussian process models on image datasets. Additionally, we show how the marginal likelihood can be used to assess to what extent a dataset can be explained with only convolutional structure. We show that convolutional structure is not sufficient, and that performance can be improved by adding a small amount of "fully connected" (RBF). The ability to do this, and automatically tune the hyperparameters is a real strength of Gaussian processes. It would be great if this ability could be incorporated in larger or deeper models as well.

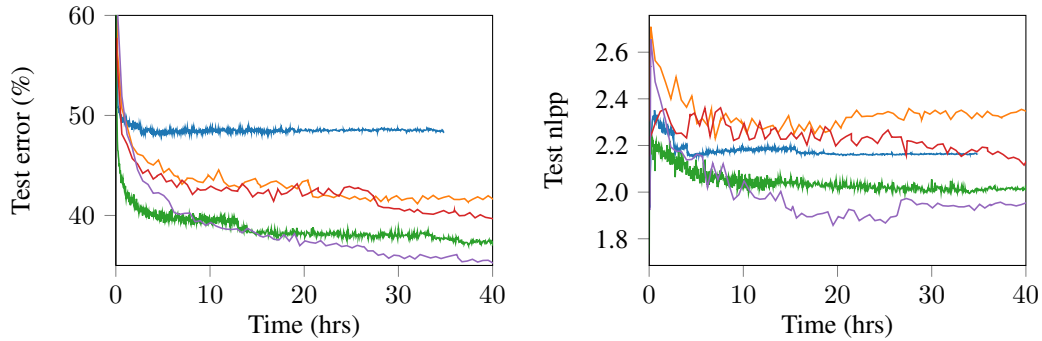

Figure 3: Test error (left) and nlpp (right) for CIFAR-10, using RBF (blue), baseline weighted convolutional (orange), full-colour weighted convolutional (green), additive (red), and multi-channel (purple).

**Acknowledgements**

CER gratefully acknowledges support from EPSRC grant EP/J012300. MvdW is generously supported by a Qualcomm Innovation Fellowship.

## Footnotes

[1]The construction of the approximate posterior can alternatively be seen as a GP posterior to a regression problem, where the $q(\mathbf{u})$ indirectly specifies the likelihood. Variational inference will then adjust the inputs and likelihood of this regression problem to make the approximation close to the true posterior in KL divergence.

[2]Ours can be found on `https://github.com/markvdw/convgp`, together with code for replicating the experiments, and trained models. It is based on GPflow [30], allowing utilisation of GPUs.

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
