[Supplementary Material]

# Convolutional Gaussian Processes: Appendix

**Mark van der Wilk**
Department of Engineering
University of Cambridge, UK
mv310@cam.ac.uk

**Carl Edward Rasmussen**
Department of Engineering
University of Cambridge, UK
cer54@cam.ac.uk

**James Hensman**
prowler.io
Cambridge, UK
james@prowler.io

## A  Implementation

### A.1  Inter-domain approximations

Implementation of convolutional kernels requires only straightforward modifications of existing code that implements an inducing variable GP approximation. All inducing point methods like FITC [1], variations on (Power-) EP [2] or variational free energy [3, 4] rely on the covariance of inducing variables and between observations and inducing variables:

$$\mathbf{K_{uu}} = \mathrm{Cov}\left[\mathbf{uu}^{\mathsf{T}}\right], \tag{A.1}$$

$$\mathbf{K_{fu}} = \mathrm{Cov}\left[\mathbf{fu}^{\mathsf{T}}\right]. \tag{A.2}$$

In normal inducing point approximations, $\mathbf{u}$ and $\mathbf{f}$ are simply evaluations of the latent GP of interest:

$$[\mathbf{f}]_n = f_n = f(\mathbf{x}_n), \qquad\qquad [\mathbf{u}]_m = u_m = f(\mathbf{z}_m). \tag{A.3}$$

The resulting covariances are simply evaluations of the kernel of the GP prior $k(\mathbf{x}, \mathbf{x}')$. Inter-domain approximations [5] simply result in a different expression for elements of $\mathbf{K_{uu}}$ and $\mathbf{K_{fu}}$, and so only require a modification to the evaluation of these matrices – the computation of the rest of the learning objective remains unchanged. As shown in the main text, our proposed inference method is an inter-domain method, and therefore can be implemented with the same small modification. The advantage of this is that all changes to the inference can be encapsulated in the kernel.

### A.2  Exploiting convolutions

A large bottleneck for the implementation is summation of kernel evaluations over numerous patches. A general implementation could simply extract the patches from the image, compute the kernel, and sum:

$$[\mathbf{K_{fu}}]_{nm} = k_{fu}(\mathbf{x}_n, \mathbf{z}_m) = \sum_p k_g(\mathbf{x}_n^{[p]}, \mathbf{z}_m). \tag{A.4}$$

This can be implemented as evaluating a large $PN \times M$ kernel matrix, reshaping to $P \times N \times M$, and summing over the first dimension. For general kernels, this is required. However, if the kernel is stationary, i.e. $k_g(\mathbf{p}, \mathbf{p}') = k_g(|\mathbf{p} - \mathbf{p}'|)$, the first step is computing the matrix of pairwise distances between all patches and inducing points. For general inputs this still doesn't help, but in this case neighbouring inputs overlap strongly, since they're all patches from the same image. By expanding the euclidean distance as

$$(\mathbf{x}_n^{[p]} - \mathbf{z}_m)^2 = \mathbf{x}_n^{[p]\mathsf{T}}\mathbf{x}_n^{[p]} - 2\mathbf{x}_n^{[p]\mathsf{T}}\mathbf{z}_m + \mathbf{z}_m^{\mathsf{T}}\mathbf{z}_m \tag{A.5}$$

we see that the inner product of $\mathbf{z}_m$ along all patches of $\mathbf{x}$ is a convolution operation (figure 1). Additionally, the inner product of the patches with themselves is also a convolution of the squared image with a window of ones. This allows the euclidean distance to be computed in $\mathcal{O}\left(\log E\right)$ rather than $\mathcal{O}\left(E\right)$. An additional speed benefit comes from being able to leverage the highly optimised code for convolutions on GPUs that was developed in light of the popularity of convnets.

Figure 1: Pictorial representation of convolution required in stationary kernels. A single dot product with a patch is highlighted.

# B   The convolutional GP can not learn any function, but is nonparametric

Here we show that there are sets of functions that a convolutional GP places no probability mass on. As a consequence, these functions will also never have density in the posterior, and can not be learned, regardless of the amount of observed data. This observation is particularly interesting, as the kernel still defines a nonparametric model. This places convolutional kernels in an interesting middle-ground, not seen in common examples of kernels. This constraint helps the convolutional kernel to generalise well, at the expense of possibly leaving some signal unexplained. However, as discussed in the main text, it is possible to get the best of both worlds by using a sum of a convolutional kernel and a kernel which spreads its probability mass more widely, and letting the marginal likelihood determine their relative weighting.

## B.1   Background: nonparametric models and consistency

Nonparametric methods are usually justified by a desire to build a method that is *universally consistent*, by which is meant that an optimal solution is found in the limit of infinite data. For example, in regression the unknown function may be any continuous function, and we would like our solution to be able to come arbitrarily close. Achieving this would require, at least, being able to represent any continuous function arbitrarily closely (the *universal approximation property*), and for Bayesian models, a prior that places probability mass over this entire space [6].

Neural networks have been shown to be able to approximate any function arbitrary closely in the limit of having an infinite number of basis functions (i.e. hidden units) [7, 8]. Kernel methods such as Gaussian processes, kernel regression or SVMs all implicitly use basis functions, possibly infinitely many, through their kernel. We can find a representation of the basis functions they implicitly use through Mercer's theorem [9, 10, 11], which represents a kernel in terms of its eigenvalues and orthogonal eigenfunctions:

$$k(\mathbf{x}, \mathbf{x}') = \sum_{i=1}^{\infty} \lambda_i \phi_i(\mathbf{x}) \phi_i^*(\mathbf{x}') \,. \tag{B.6}$$

The span of the functions $\left\{ \sqrt{\lambda_i} \phi_i \right\}_i$ determines exactly the functions that a kernel method can capture. For SVMs and kernel regression, the estimated function lies in the RKHS spanned by these bases, while the sample paths of a Gaussian process can be constructed as the infinite sum of eigenfunctions weighted by Gaussian random variables [10, §5.1]:

$$f(\cdot) = \sum_{i=1}^{\infty} w_i \phi_i(\cdot) \,, \qquad\qquad w_i \sim \mathcal{N}\left(0, \lambda_i\right) \,. \tag{B.7}$$

**Degenerate kernels** We call any kernel with only a finite number of non-zero eigenvalues *degenerate* (following [11]). Degenerate kernels can not lead to models that are universal approximators, as any function that contains a component of an eigenfunction with a zero eigenvalue will be outside the RKHS and outside the set of functions the GP can generate. Furthermore, such a function can have an arbitrarily large deviation from functions that can be represented, simply by adding a larger component of the eigenfunction with zero eigenvalue. From this, we can easily see that an infinite number of non-zero eigenvalues is necessary for universal approximation.

We can alternatively see this by considering that a degenerate kernel with $B$ non-zero eigenvalues can be expressed as a finite basis function model. In these models, we can fully specify any function with knowledge of $B$ function values. The function value at a $B + 1$th input is therefore also fully constrained. We can construct a function outside the RKHS or prior simply by adding a perturbation to the constrained function which does not pass through the $B + 1$th output.

Common degenerate kernels arise from considering parametric models, like linear or polynomial models.

**Universal kernels** Steinwart [12, def. 4] introduced the concept of a *universal* kernel, which has an RKHS which is dense in the space of all continuous functions, i.e. for every continuous function, there is a function in the RKHS with an arbitrarily small maximum deviation. Gaussian processes based on universal kernels have sample paths which are arbitrarily close to any continuous function[1]. The universal consistency arguments for SVM classification and kernel regression by Steinwart [14] and Christmann and Steinwart [15] rely on using universal kernels. Micchelli et al. [16] further characterise the properties required for universal kernels. Most common non-degenerate kernels, like the squared exponential, are also universal [16].

## B.2 Convolutional kernels: nonparametric but not universal

Here, we show that convolutional kernels fall between degenerate and universal kernels in terms of their representational capacity. We first show that we can construct a collection of inputs which fully constrains the function value at a different input, as was discussed for degenerate kernels. We then follow on to show that unlike degenerate kernels, we can still arbitrarily specify the function at an infinite number of distinct points, showing that the kernel can not implicitly be using a finite number of basis functions.

**Claim.** *Weighted covariance kernels are not universal, and Gaussian processes based on them do not place probability on (or near) all continuous functions.*

*Proof.* Consider $W \times H$ sized images with $w \times h$ sized patches. If $w < W$ and $h < H$, we will have $P > 1$ patches in each image. There are $WH$ images with a single pixel switched on, and $wh$ distinct patches $\{\mathbf{z}_i\}_{i=1}^{wh}$. We can organise the evaluations of $g(\cdot)$ for each of the $wh$ distinct patches in the vector $\mathbf{g} \in \mathbb{R}^{wh}$, where $[\mathbf{g}]_i = g(\mathbf{z}_i)$. If we consider $N \leq WH$ image inputs with a single pixel switched on, we can obtain the function values $\mathbf{f} \in \mathbb{R}^N$ through the linear transformation

$$\mathbf{f} = WQ\mathbf{g}\,. \tag{B.8}$$

$W \in \mathbb{R}^{N \times P}$ has the patch weights as rows, and the matrix $Q \in \mathbb{R}^{P \times wh}$ contains a 1 at $Q_{ni}$ when the $n$th image contains the patch $i$, and zero elsewhere. The matrix $WQ$ has size $N \times wh$. This implies that for $N = wh + 1$, one of the function values in $\mathbf{f}$ will be fully determined by the responses of the previous images. As a consequence, the kernel matrix for these inputs has to have some zero eigenvalues because the matrix

$$\mathbb{E}\left[\mathbf{f}\mathbf{f}^\mathsf{T}\right] = \mathbf{K_{ff}} = \mathbb{E}\left[WQ\mathbf{g}\mathbf{g}^\mathsf{T}W^\mathsf{T}Q^\mathsf{T}\right] = WQK_G Q^\mathsf{T}W^\mathsf{T} \tag{B.9}$$

has rank at most $wh$, which shows that all functions with evaluations $\mathbf{f}$ with a component in the null space of $\mathbf{K_{ff}}$ have no density under the prior.

The construction of a singular kernel matrix $\mathbf{K_{ff}}$ also implies that the kernel is not strictly positive definite, and therefore not universal [17]. □

**Claim.** *Convolutional kernels are nonparametric, in that they can not be represented as a finite basis function model.*

*Proof.* If the kernel only had a finite number of non-zero eigenvalues and the model could be expressed as a finite basis function model, all functions would admit the representation:

$$f(\mathbf{x}) = \phi(\mathbf{x})^{\mathsf{T}}\mathbf{w}. \tag{B.10}$$

where $\phi : \mathbb{R}^D \to \mathbb{R}^I$. The corresponding kernel matrix would have at most $\operatorname{rank} I$.

We choose $N$ distinct images by with a distinct patch in the top left corner of the image of size $w \times h$, all other pixels being zero. Because patches overlap, we get $wh$ distinct patch responses which influence $f(\mathbf{x})$ as well as the influence from all the zero patches:

$$f(\mathbf{x}) = \sum_i g(\mathbf{x}^{[i]})w_i + w_0 g(\mathbf{0}) \tag{B.11}$$

We collect the patch responses in $G \in \mathbb{R}^{N \times wh}$, with the weights $\mathbf{w} \in \mathbb{R}^{wh}$, with the image evaluations becoming $\mathbf{f} = G\mathbf{w}$. We obtain the covariance of $\mathbf{f}$:

$$[\mathbf{K_{ff}}]_{nn'} = \mathbb{E}_g\left[ \sum_{i=1}^{wh}\sum_{j=1}^{wh} g\left(\mathbf{x}_n^{[i]}\right) g\left(\mathbf{x}_{n'}^{[j]}\right) w_i w_j \right] = \sum_{i=1}^{wh}\sum_{j=1}^{wh} k_g\left(\mathbf{x}_n^{[i]}, \mathbf{x}_{n'}^{[j]}\right) w_i w_j. \tag{B.12}$$

This covariance matrix can be obtained by reducing down the $Nwh \times Nwh$ covariance matrix between all patches. If we choose a universal kernel for $k_g(\cdot, \cdot)$, this matrix will always be positive definite. The reduced matrix is also positive definite since:

$$\mathbf{a}^{\mathsf{T}}\mathbf{K_{ff}}\mathbf{a} = \sum_{nn'} a_n a_{n'} \sum_{i=1}^{wh}\sum_{j=1}^{wh} [\mathbf{K_{gg}}]_{nin'j}\, w_i w_j = \sum_{nin'j} [\mathbf{K_{gg}}]_{nin'j}\, a_n w_i a_{n'} w_j > 0 \tag{B.13}$$

This contradicts the model being parametric, which would allow the rank of $\mathbf{K_{ff}}$ to be at most $I$. □

## B.3   Remarks

The existence of non-degenerate kernels which are not universal may not come as a surprise to theoreticians, particularly due to the effort required for proving universality. For example, Micchelli et al. [16] place strong requirements on the form of the implicit features of the kernel, which are likely not satisfied by convolutional kernels. Despite the prescience of theory, convolutional kernels provide an interesting and practically useful example of such kernels.

## C   Variational bound for separate representation of latent GPs

In the main text (sections 3.3 & 3.4) we saw two examples of models with additive structure that required separate representation of their inducing outputs. The weighted convolution + RBF experiment required this due to the inducing inputs lying in separate spaces, while the multi-channel convolutional kernel required this due to separate inter-domain inducing outputs being required to find the distribution over the GP output. In both cases, we can construct the GP output of interest from components in the same space as the inducing variables:

$$f_{sum}(\mathbf{x}) = f_{rbf}(\mathbf{x}) + \sum_p w_p g(\mathbf{x}^{[p]}) \qquad f_{multi}(\mathbf{x}) = \sum_{p=1}^{P}\sum_{c=1}^{C} w_{pc} g_c\left(\mathbf{x}^{[pc]}\right) \tag{C.14}$$

$$f_{rbf}(\cdot) \sim \mathcal{GP}\left(0, k_{rbf}(\cdot, \cdot)\right) \qquad\qquad g_c(\cdot) \sim \mathcal{GP}\left(0, k_g(\cdot, \cdot)\right) \tag{C.15}$$

$$g(\cdot) \sim \mathcal{GP}\left(0, k_g(\cdot, \cdot)\right) \tag{C.16}$$

Here we show in detail how inference is done in these models, and how no expensive operations are performed on matrices larger than $M \times M$.

## C.1 Defining the inducing variables

We first choose our inducing variables. For the summed GP, we choose $M$ evaluations of $f_{rbf}(\cdot)$ and $g(\cdot)$ each (giving $2M$ inducing variables), while for the multi-channel GP we choose $M$ evaluations of all $C$ colour channel outputs $g_c(\cdot)$ (giving $MC$ inducing variables). We can construct the sparse approximate posterior by conditioning the prior on these variables. The form of the posterior is exactly the same as usual:

$$f(\cdot)|\mathbf{u} \sim \mathcal{GP}\left(\mathbf{k_u}(\cdot)^{\mathsf{T}}\mathbf{K_{uu}^{-1}}\mathbf{u}, k(\cdot,\cdot) - \mathbf{k_u}(\cdot)^{\mathsf{T}}\mathbf{K_{uu}^{-1}}\mathbf{k_u}(\cdot)\right) . \tag{C.17}$$

As with usual inter-domain approximations, the task is to find the correct covariances, again by finding the covariances from equations (10) and (11).

**Summed kernels** For $f_{sum}(\cdot)$, the cross-covariance $k_{fu}(\mathbf{x}, \mathbf{z})$ will be the regular kernel evaluation for inducing points on $f_{rbf}(\cdot)$, and the appropriate cross-covariance for the inducing patches. We order the inducing point covariances above the inducing patch covariances in the matrix $\mathbf{K_{fu}}$. Additionally, $\mathbf{K_{uu}}$ will be a block-diagonal $2M \times 2M$ matrix, with the inducing point $M \times M$ matrix for $f_{rbf}(\cdot)$ in the top left, and the inducing patch covariances in the bottom right. No cross terms between $f_{rbf}(\cdot)$ and $g(\cdot)$ appear, as they are independent in the prior.

$$k(\mathbf{x}, \mathbf{z}_{img}) = \mathbb{E}_{f_{rbf},g}\left[\left(f_{rbf}(\mathbf{x}) + \sum_p w_p g(\mathbf{x}^{[p]})\right) f_{rbf}(\mathbf{z}_{img})\right] = k_{rbf}(\mathbf{x}, \mathbf{z}_{img}) \quad \text{(C.18)}$$

$$k(\mathbf{x}, \mathbf{z}_{patch}) = \mathbb{E}_{f_{rbf},g}\left[\left(f_{rbf}(\mathbf{x}) + \sum_p w_p g(\mathbf{x}^{[p]})\right) g(\mathbf{z}_{patch})\right] = k_g(\mathbf{x}, \mathbf{z}_{patch}) \quad \text{(C.19)}$$

$$k(\mathbf{z}_{img}, \mathbf{z}_{patch}) = \mathbb{E}_{f_{rbf},g}\left[f_{rbf}(\mathbf{z}_{img})g(\mathbf{z}_{patch})\right] = 0 \tag{C.20}$$

**Multi-channel kernels** For $f_{multi}(\cdot)$ the situation is similar, with the difference that we only have $M$ inducing inputs, but $MC$ inducing outputs. If we order the inducing variables by colour, we get an $N \times MC$ $\mathbf{K_{fu}}$ matrix (as in equation (21)), and a block-diagonal $\mathbf{K_{uu}}$, as:

$$\mathbb{E}_{\{g_c\}_{c=1}^C}\left[g_c(\mathbf{z})g_{c'}(\mathbf{z}')\right] = k_g(\mathbf{z}, \mathbf{z}')\delta_{cc'} . \tag{C.21}$$

This process is slightly different compared to usual inducing variable approximations, and the even the case for summed kernels, as the number of inducing variables is larger than the number of inducing inputs. As a curiosity, and not necessarily a practical method of implementation, we would like to point out that we could view this process as having multi-output inducing variables. The function $g : \mathbb{R}^{wh} \to \mathbb{R}^C$ could collect all $g_c(\cdot)$s, as one $\mathbb{R}^C$ variable.

## C.2 Inference with block-diagonal $\mathbf{K_{uu}}$ matrices

In the previous section, we saw how to find the conditional process. Here we show that the marginal likelihood bound

$$\text{ELBO} = \sum_i \mathbb{E}_{q(f(\mathbf{x}_i))}\left[\log p(y_i \,|\, f(\mathbf{x}_i))\right] - \text{KL}[q(\mathbf{u})||p(\mathbf{u})] \tag{C.22}$$

can be computed without operations on matrices larger than $M \times M$, despite using more than $M$ inducing variables, regardless of the mean-field assumptions between inducing variables.

**Approximate posterior marginals** The bound requires computation of the marginals of the approximate posterior $q(f(\mathbf{x}_i))$. This requires marginalising the conditional approximate posterior over $\mathbf{u}$ (the same procedure as in [4, 18]). This is where equations (6-8) come from. We simply substitute in the $\mathbf{K_{fu}}$ and $\mathbf{K_{uu}}$ matrices from the previous section, and simplify using the block-diagonal structure in $\mathbf{K_{uu}}$. We refer to each group of inducing variables ($C$ in total) that are correlated in a block of $\mathbf{K_{uu}}$ as $\mathbf{u}_c$, and the corresponding covariance matrices $\mathbf{K_{u_c u_{c'}}}$ and $\mathbf{k_{u_c}}(\mathbf{x})$. We similarly split the variational parameters $\mathbf{m}$ and $\mathbf{S}$ into blocks of the same size $\mathbf{m}_c$ and $\mathbf{S}_{cc'}$.

$$\mu_i = \mathbf{k_u}(\mathbf{x})^{\mathsf{T}}\mathbf{K_{uu}^{-1}}\mathbf{m} = \sum_c \mathbf{k_{u_c}}(\mathbf{x})^{\mathsf{T}}\mathbf{K_{u_c u_c}^{-1}}\mathbf{m}_c \tag{C.23}$$

$$\sigma_i^2 = k(\mathbf{x}_i, \mathbf{x}_i) + \mathbf{k_u^{\mathsf{T}}}(\mathbf{x})\mathbf{K_{uu}^{-1}}(\mathbf{S} - \mathbf{K_{uu}})\mathbf{K_{uu}^{-1}}\mathbf{k_u}(\mathbf{x})$$

$$= k(\mathbf{x}_i, \mathbf{x}_i) + \sum_{cc'} \mathbf{k_{u_c}}(\mathbf{x})^{\mathsf{T}}\mathbf{K_{u_c u_{c'}}^{-1}}\mathbf{S}_{cc'}\mathbf{K_{u_c u_{c'}}^{-1}}\mathbf{k_{u_c}}(\mathbf{x}) + \sum_c \mathbf{k_{u_c}}(\mathbf{x})^{\mathsf{T}}\mathbf{K_{u_c u_c}^{-1}}\mathbf{k_{u_c}}(\mathbf{x}) \quad \text{(C.24)}$$

In both cases outlined above, $\mathbf{K}_{\mathbf{u}_c \mathbf{u}_{c'}}$ is $M \times M$. If a mean-field approximation is chosen $\mathbf{S}_{cc'} = 0$ when $c \neq c'$. This does not impact the number or size of any inverses, only requiring less parameters and avoiding a summation over $c'$.

**KL divergence**    The second term in the bound requires the KL divergence between the prior and posterior distribution over the inducing variables, which we can again simplify using knowledge of the block-diagonal structure.

$$\mathrm{KL}\left(q(\mathbf{u})\|p(\mathbf{u})\right) = \frac{1}{2}\left(\mathrm{Tr}\left(\mathbf{K}_{\mathbf{uu}}^{-1}\mathbf{S}\right) + \boldsymbol{\mu}^{\mathsf{T}}\mathbf{K}_{\mathbf{uu}}^{-1}\boldsymbol{\mu} - MC + \log\frac{|\mathbf{K}_{\mathbf{uu}}|}{|\mathbf{S}|}\right) \tag{C.25}$$

$$= \frac{1}{2}\left(\sum_c \mathrm{Tr}\left(\mathbf{K}_{\mathbf{u}_c\mathbf{u}_c}^{-1}\mathbf{S}_{cc}\right) + \sum_c \boldsymbol{\mu}_c^{\mathsf{T}}\mathbf{K}_{\mathbf{u}_c\mathbf{u}_c}^{-1}\boldsymbol{\mu}_c - MC + \sum_c \log|\mathbf{K}_{\mathbf{u}_c\mathbf{u}_c}| - \log|\mathbf{S}|\right) \tag{C.26}$$

Now, the determinant of $\mathbf{S}$, which may be of size $MC \times MC$ remains. Luckily, we are free to choose the parameterisation of this matrix. We parameterise this matrix as $\mathbf{S} = LL^{\mathsf{T}}$, which makes $\log|\mathbf{S}| = 2\sum \log \mathrm{diag}L$.

### C.3    Summary

Here we showed that when the prior covariance of the inducing outputs is block-diagonal, the inference only requires expensive matrix operations on each of the blocks separately, regardless of the posterior correlations taken into account. This allows efficient inference for the summed and multi-channel convolutional kernels considered here.

## D    Inter-domain inducing variables for general invariances

We finally briefly show that the inter-domain trick used for convolutional kernels can also be applied to kernels that give rise to Gaussian processes with arbitrary invariances. Invariant kernels have been discussed before, notably by Kondor [19] and Ginsbourger et al. [20, 21, 22]. Duvenaud [23, 24] also provides an accessible discussion. Here, we review the connection between kernels resulting in invariant functions and a summation structure which allows our inter-domain trick to be applied.

### D.1    Specifying invariances in kernels

An invariance can be formalised by placing equality constraints on $f(\cdot)$ under transformations of the input. Consider a collection of transformations from the input space to itself $g_i : \mathcal{X} \to \mathcal{X}$. Making $f(\cdot)$ invariant to these transformations specifies that

$$f(\mathbf{x}) = f(g_i(\mathbf{x})) \qquad\qquad \forall \mathbf{x} \in \mathcal{X} \qquad\qquad \forall i \,. \tag{D.27}$$

Kondor [19] and Ginsbourger et al. [20] discuss that this requirement is equivalent to invariance under every composition of transformations as well. For example, if $g_1(\cdot)$ and $g_2(\cdot)$ are translations upwards and to the right respectively, we must also have invariance to a translation up and to the right $f(\mathbf{x}) = f(g_1(g_2(\mathbf{x})))$. The set of compositions of all transformations forms a group $G$. Ginsbourger et al. [20, theorem 3.1] show that in order for samples $f(\cdot)$ to be invariant to all compositions of transformations, the kernel must be *argumentwise invariant*:

$$k(\mathbf{x}, \mathbf{x}') = k(g(\mathbf{x}), g'(\mathbf{x}')) \qquad\qquad \forall \mathbf{x}, \mathbf{x}' \in \mathcal{X} \qquad\qquad g, g' \in G \,. \tag{D.28}$$

The elements of the group $g \in G$ are all compositions of the transformations $g_i$, defined above.

### D.2    Constructing invariant kernels

The requirement stated above does not directly help with constructing invariant models. Three main methods have been proposed, which are neatly discussed by Duvenaud [23, §2.7]. For our purposes, we are mainly interested in the "summation over orbit" method, as this gives a structure almost identical to the convolutional kernel.

Kondor [19] and Ginsbourger et al. [20] show that an argumentwise invariant kernel can be constructed by summing some base kernel over the *orbits* of $\mathbf{x}$ and $\mathbf{x}'$. The orbit of a point $\mathbf{x}$ with respect to

a group $G$ is defined as the set of all points obtained from applying each element of $G$ to $\mathbf{x}$: $\mathcal{O}_G(\mathbf{x}) = \{g(\mathbf{x}) \mid g \in G\}$. The resulting kernel becomes:

$$k_{invariant}(\mathbf{x}, \mathbf{x}') = \sum_{\tilde{\mathbf{x}} \in \mathcal{O}_G(\mathbf{x})} \sum_{\tilde{\mathbf{x}}' \in \mathcal{O}_G(\mathbf{x}')} k_{base}(\tilde{\mathbf{x}}, \tilde{\mathbf{x}}') . \qquad (D.29)$$

The relation between invariances and the addition structure is further investigated by Ginsbourger et al. [25].

Kernels constructed in this way have the same computational issues as convolutional kernels: evaluating the kernel for a single pair of points requires $P^2$ base kernel evaluations, where $P$ is the size of the orbit. For example, we could make a fully translation invariant kernel by considering translations by 1 pixel upwards, downwards and to the left and right, while clipping and zero-padding edges. For images of size $W \times H$ the orbit would consist of all $W - 1 \times H - 1$ translated images. For MNIST this would give $P^2 = (27 \times 27)^2 = 5.3 \cdot 10^5$, which is again impractical.

### D.3 Inter-domain inducing variables for invariant kernels

The invariant kernel above can also be obtained by considering a model that sums a base function $f_{base}(\mathbf{x}) \sim \mathcal{GP}(0, k_{base}(\cdot, \cdot))$ over the orbit of $\mathbf{x}$:

$$f(\mathbf{x}) = \sum_{\tilde{\mathbf{x}} \in \mathcal{O}_G(\mathbf{x})} f_{base}(\tilde{\mathbf{x}}) . \qquad (D.30)$$

In this construction, the base function $f_{base}(\cdot)$ takes the place of the patch response function $g(\cdot)$ from the convolutional kernel, allowing us to use the same inter-domain trick. Instead of using normal inducing inputs, we place the inducing inputs in $f_{base}(\cdot)$ instead. We then obtain the covariances:

$$k(\mathbf{x}, \mathbf{z}) = \sum_{\tilde{\mathbf{x}} \in \mathcal{O}_G(\mathbf{x})} k_{base}(\tilde{\mathbf{x}}, \mathbf{z}) , \qquad (D.31)$$

$$k(\mathbf{z}, \mathbf{z}') = k_{base}(\mathbf{z}, \mathbf{z}') . \qquad (D.32)$$

Just like with the convolutional kernel, this reduces the cost of evaluating the required kernels significantly.

### D.4 Summary

The structure of kernels resulting in GPs that are invariant to specified transformations is almost identical to that of convolutional kernels, allowing the same inter-domain trick to be used to speed up inference. We present the derivation here, but leave empirical demonstration and evaluation to future work.

## Footnotes

[1]This follows from Ghosal and Roy [13, theorem 4], which shows that Gaussian processes assign non-zero probability to functions that are close to functions in the RKHS of their kernel.