[Reviews · NeurIPS 2017]

Reviewer 1



The paper presents a method to incorporate convolutional structure in GPs, in order to make them more competitive for image classification problems. This is achieved by constructing a convolutional kernel through a latent function g() shared by different patches in the image. Approximate inference is based on variational methods and very promising results are given in MNIST and CIFAR datasets. The paper is clearly written and it presents a quite extensive set of experimental results. Regarding the main construction of the convolutional kernel (see eq. 13 and 14) I am not sure if the concept of inter-domain inducing variables is really necessary or useful here. The latent function of the GP model is essentially g() which is used inside the likelihood through this patch-sharing, i.e. through the definition of f(x). Under this viewpoint the inducing variables are just points of the latent function, i.e. points of g(), and you don't really need to refer to inter-domains in order to describe the variational sparse approximation. Also having presented the method based on g() (and discussing f(x) only as a complementary viewpoint) will probably make it easier for the reader to understand the weighted and colour extensions. Can you clarify further the arguments in lines 215-220? In particular, it is not clear why you avoid the inversion of a 2 M × 2 M matrix given that q(u1,u2) is coupled. Section 3 in Supplementary material, that tries to clarify this point, is very unclear and it contains also mistakes (e.g. in equation 13). Overall I believe that this is a good paper since it improves a lot on GP classification by incorporating convolutional structure. Still of course GPs cannot compete with Deep NNs, but this paper introduces some new ideas and it makes good steps forward to making GP classifiers more competitive for image classification.

Reviewer 2



The authors propose an approach to construct convolutional kernel functions for Gaussian processes. That way Gaussian process classification can learn and use non-local features similar to a convolutional neural network. Results for image classification show that convolutional kernels have advantages compared to a standard RBF kernel. The paper is well written and shows both the strengths and weaknesses of the approach. The construction of the convolutional kernels is documented clearly. And section 2.1 contains a good description of the inference method used for the experiments. The original contribution of this paper is the method for constructing convolutional kernels based on patch response function modeled as Gaussian process. As this approach enables non-local features for Gaussian process classification, I expect that it will be used widely and developed further. I have read the author feedback. In my opinion presenting convolutional Gaussian processes as "just another kernel" is a good choice, as algorithms and software frameworks for inference with Gaussian processes typically can use custom kernels but only a limited set of likelihood functions.

Reviewer 3



This paper proposes a new formulation that provides Gaussian processes with convolutional nature, aiming at exploiting some of the advantages of this type of models, so well-known in deep neural networks. The contribution is technically solid, and a worthy contribution to the literature and toolbox of Gaussian processes. However, the paper is not very clearly written, as it seems to have been written with an expert on Gaussian processes as the target reader. The experimental results are a bit disappointing, especially those on the CIFAR-10 dataset.